

# An overview of biomass conversion: exploring new opportunities

László Fülöp and János Ecker

Department of Chemistry, Szent István University, Gödöllő, Hungary

## ABSTRACT

Recycling biomass is indispensable these days not only because fossil energy sources are gradually depleted, but also because pollution of the environment, caused by the increasing use of energy, must be reduced. This article intends to overview the results of plant biomass processing methods that are currently in use. Our aim was also to review published methods that are not currently in use. It is intended to explore the possibilities of new methods and enzymes to be used in biomass recycling. The results of this overview are perplexing in almost every area. Advances have been made in the pre-treatment of biomass and in the diversity and applications of the enzymes utilized. Based on molecular modeling, very little progress has been made in the modification of existing enzymes for altered function and adaptation for the environmental conditions during the processing of biomass. There are hardly any publications in which molecular modeling techniques are used to improve enzyme function and to adapt enzymes to various environmental conditions. Our view is that using modern computational, biochemical, and biotechnological methods would enable the purposeful design of enzymes that are more efficient and suitable for biomass processing.

## INTRODUCTION

The importance of biomass recycling is that the usage of fossil fuels can be reduced towards the goal of realizing a more sustainable mix of energy sources. Biomass is a renewable energy source, and is reproduced in a short life cycle, usually within 1 year (*Wang et al., 2018*). Plant biomass that is recycled annually on Earth is estimated to be about 200 billion ($2 \times 10^{11}$) tonnes/year (*Bjerre et al., 1996*). Most of this is in the form of lignocellulose. The aim of biomass recycling is to extract, process and convert the major carbohydrate components (sucrose, starch, cellulose, hemicellulose) in plant material into valuable chemical components or fuel (*Schlaf & Zhang, 2016*). With the use of biomass, the use of current energy sources (coal, natural gas, oil) can be reduced. This can result in lower air pollution and slow down the rate of increase in the athmospheric $CO_2$ content (suppressing the greenhouse effect and global warming) (*Song, Wang & Tang, 2016*). We believe that it is important to explore all opportunities to apply existing physical,

Corresponding author
László Fülöp,
Dr.Fulop.Laszlo@gmail.com

chemical, microbiological, biochemical and biotechnological knowledge, and apply the results to improve biomass conversion and utilization processes.

## Plant cell wall polysaccharides

Cellulose is one of the most stable molecules on Earth. The glucose associated with the β-1,4-O-glucosidic bonds forms cellulose (*Heinze, Seoud & Koschella, 2018*). Within the polymer molecule, H-bonds are formed between the ring's oxygen and the 3-OH group, while H-bond is also formed through the O-glycosidic bond and the 6-OH group between the chains (*Shuangxi et al., 2020*). This cross-linked structure effectively prevents the enzymatic degradation of the polymer molecule since the O-glycosidic bonds are difficult to access for enzymes (*Rosenau, Potthast & Hell, 2018*). Thus, the hydrolysis of cellulose molecules is difficult to accomplish without help. The breakdown of cellulose is also hampered by the fact that hydrolysis of the O-glycosidic bonds may be reversed due to the immobility of the molecules.

Hemicelluloses are polysaccharides that, in addition to cellulose, are involved in building up the cell wall (*Houfani et al., 2020*). Their role is to fix the cell wall skeleton and connect it to the cellulose fiber network. The practical distinction between cellulose and hemicellulose is made possible by the fact that cellulose is not soluble in NaOH solution, while polysaccharides with lower molecular weight and the different polysaccharides of hemicellulose are soluble (*Kojima et al., 2019*). On the other hand, acid hydrolysis of hemicelluloses takes place under milder conditions than that of cellulose; with dilute sulfuric acid, almost all of the hemicellulose content can be liquefied into monomers without additional enzymatic or acid treatment. Cellulose can only be hydrolyzed with concentrated hydrochloric acid or 60–80% sulfuric acid at more than 100 °C temperature and under high pressure (*Lee & Yu, 2020*).

The hemicellulose content of plants is predominantly pentosan, and thus arabinan is formed by L-arabinose associated with 1,2-, 1,3- and 1,5-O-glycosidic bonds. Xylan consists of xylose linked by beta-1,4-O-glycosidic bonds. The chain may be linked to alpha-L-arabinose and alpha-D-glucose (*Serna-Saldívar & Hernández, 2020*). Xyloglucan and arabino-galactan also contain sugars composed of five and six carbon atoms. Galactomannans are made up of beta-D-mannoses which are connected by beta-1,4-O-glycosidic bonds, formed by a side chain bound by beta-D-galactose with 1,6-O-glycosidic bonds (*Rodriguez-Canto et al., 2019*).

Pectins, together with hemicelluloses and proteins, form the base matrix of the middle plate and cell walls (*Stoffels et al., 2020*). Most of the pectin consists of alpha-D-galacturonic acid. The chain is interrupted in part by alpha-L-rhamnose, to which the alpha-L-arabinose chain and the alpha-D-galactose chain may be attached (*Shakhmatov, Toukach & Makarova, 2020*).

Lignin is deposited in the molecular size cavities of the cell wall of cellulose and hemicellulose, increasing the mechanical resistance of the cell wall (*Setälä et al., 2020*). Lignin found in different wood species or in plant parts have different structure, but there are structural similarities: the aromatic rings are often linked by a methoxy group or a phenolic hydroxyl group (*Li et al., 2020*). Lignin is a constituent of tissues of plants.

Three cinnamon alcohol derivatives (sinapyl alcohol, *p*-coumaryl alcohol, coniferyl alcohol) are involved in the construction of lignin. From these, the macromolecular lignin is formed by the action of a dehydrogenase enzyme (*Elder et al., 2020*). Lignin can interact in several ways with cellulose and hemicellulose.

### Review methodology

The following search terms were used for database search: polysaccharides, cellulose hemicellulose lignin biodegradation, modification, immobilization, industry. *O*-glucosyl hydrolase, cellulase enzyme modeling, molecular modeling, computation, cellulase modification, cellulose enzyme modification modeling. The search was combined with OR and AND terms. Searches were carried out up until 2020. We have also reviewed the articles and references therein. The articles we found interesting with respect to the problem area highlighted above were classified into the following categories: Biodegradation of polysaccharides; Mechanical and physico-chemical modifications of substrates; Cellulose hemicellulose lignin modification; Modification of enzymes; Immobilization of enzymes; Molecular modeling of enzymes. From these articles, we compiled our study to examine the use of model-based enzyme modifications or rational design in contrast to conventional methods.

## BIODEGRADATION OF POLYSACCHARIDES

The production of ethanol, which can also be used as a fuel, is still almost exclusively fermentation-based. For this, the raw material (sucrose and starch) is obtained from sugar cane, sugar beet, cereals, and potatoes (*Arnold, Tainter & Strumsky, 2019*). These so-called first-generation biofuel technologies necessarily raise ethical, environmental and political concerns due to the raw materials used in these processes. Therefore, second-generation technologies (*Satari, Karimi & Kumar, 2019*) are under constant development. These technologies can utilize lignocellulose-containing byproducts of agriculture and forestry, as well as industrial wastes or residues along with crops that are not used for food production. Unlike the first-generation methods that process only a relatively small proportion of plants (root, stem, crop), processing the entire biomass is possible with second-generation technologies. The third-generation methods (*Mahjoub et al., 2020*) are used to process algae and microorganisms into biofuels. Fourth-generation biofuel processes use genetically modified algae to further enhance fuel production.

How can biomass-based fuel production methods be improved for greater efficiency or productivity? One option is modifying processing conditions (temperature, pH, physical and chemistry parameters) or altering the substrate with physical chemistry methods. Another possibility is to look for newer and more specific enzymes to utilize biomass. The third approach is to modify the existing enzymes and increase their efficiency. These options are reviewed below.

### Mechanical, physico-chemical and biotechnological modifications of substrates

Pre-treatment of biomass can be physical, chemical, physico-chemical and biological. The recovered polysaccharides (cellulose, hemicellulose) can be degraded by mild acidic or

enzymatic hydrolysis (*Cann et al., 2020*). Monosaccharides (glucose, mannose, xylose, arabinose, etc.) obtained by hydrolysis of polysaccharides are sources of valuable components (methanol, ethanol, furfural and derivatives, hydroxymethyl-furfural, levulinic acid, succinic acid, lactic acid, sorbitol, etc.) (*Van Walsum et al., 1996*; *Ramli & Amin, 2020*). The purpose of physical pretreatment is to reduce the particle size by grinding. The energy requirements are usually high, so these methods cannot be economical procedures on an industrial scale.

One possible way of chemical pretreatment is the lignin breakdown with ozone, which occurs at room temperature under atmospheric pressure and does not produce toxic byproducts. However, this method is very expensive due to the use of ozone (*Figueirêdo, Heeres & Deuss, 2020*). Acid or alkaline hydrolysis is more important in practice (*Kobayashi & Fukuoka, 2013*). An example of the former is mild acid pre-treatment with sulfuric acid, which can be carried out in a high-temperature continuous reactor. Hemicelluloses can be released from the plant cell wall with alkali (*Tribot et al., 2019*). After alkaline extraction, most hemicellulose fractions dissolve in water (*Teymouri et al., 2005*; *Deepa et al., 2011*; *Kamm, Gruber & Kamm, 2016*). General experience shows that treatment with ionic liquids causes both carbohydrates and lignin to be soluble but not degraded (*Ramli & Amin, 2020*). Another possibility for chemical pre-treatment is the use of organic solvents; in this case, an organic solvent is used in the presence of an inorganic acid catalyst to break down the bonds between hemicelluloses and lignin to bring the lignin into solution (*Wang et al., 2020a*). The structure of lignin and the linkages between lignin and hemicellulose and cellulose can be broken by acetyl group removal as well, and the successive enzymatic hydrolysis of cellulose can be enhanced (*Huang et al., 2019a*).

The impact of various pre-treatment processes on improving the surface morphology of wheat straw has been investigated using polystyrene composite films. The surface was improved by the enhancement of the susceptible cellulose area; therefore, the modified material can be used for various industrial green packaging applications (*Dixit & Yadav, 2019*). The effect of alkali-bleach treatment has successfully obtained microfibrillated cellulose from stalk sweet sorghum waste fibers (*Ismojo et al., 2019*). Almost complete biomass saccharification was achieved with 4% NaOH at 50 °C (*Alam et al., 2019*). A group showed that ball milling can lead to the depolymerization of cell-wall polymers, especially the polysaccharides. Micromorphology characterization showed that mechanical manipulation disintegrated fibrillar matrices. The β-1,4 glycosidic bond breakage in cellulose, along with the decomposition of arabinoxylans suggested the modification in polysaccharide chains (*Liu et al., 2019b*). In conclusion, mechanochemical methods can be used to create more digestible plant material.

The addition of naphthol derivatives can enhance the enzymatic hydrolysis, as was described by *Lai et al. (2018)* and *Fei et al. (2020)*. *P*-toluenesulfonic acid can remove wood lignin. Disk refining with subsequent acid hydrolysis (so-called physicochemical treatment) doubled the delignification efficiency (*Gu et al., 2019*).

In a 2-propanol alkaline medium, arabinoxylans were derivatized by carboxymethylation with sodium monochloroacetate. With this method, it is possible to produce hemicellulosic

derivatives from corn fibers without using extreme conditions in solvents and temperature (*De Mattos, Colodette & De Oliveira, 2019*). Lignocellulose crops serve as an excellent feedstock for biofuels because of their reduced costs and net carbon emission, and higher energy efficiency. Results showed that *Miscanthus sacchariflorus* possesses lower lignin and higher polysaccharide content in its leaves and stalks, compared to other Miscanthus species, therefore it is a better bioenergy crop (*Jung, Kim & Chung, 2015*). It was found that an alkali treatment leads to gradual removal of binding materials, such as hemicellulose and lignin from bamboo fiber (*Zhang, Wang & Keer, 2015*; *Chin et al., 2020*). The wood waste and cellulose, hemicellulose handled with cyclic anhydrides in a green reactive and solvent-free extrusion will allow targeted modification of composites (*Vaidya, Gaugler & Smith, 2016*). Suhas focused on the utilization of cellulose as an adsorbent in natural/modified form or as a precursor for activated carbon (*Suhas et al., 2016*; *Liu et al., 2020*). Cellulose, lignin, and hemicellulose were tested to hydrothermal carbonization in different temperature ranges and time. The effects of these parameters were combined and investigated in-depth (*Borrero-López et al., 2018*). Modification of pectin improves its physicochemical properties, thus allowing the availability of other components (*Buergy et al., 2020*; *Wang et al., 2020b*).

Modified microfibrillated cellulose was used with three different coupling agents: 3-aminopropyl triethoxysilane, 3-glycidoxypropyltrimethoxysilane, and titanate-containing agent to improve the mechanical properties of the cellulose-based material. The modifications changed the surface character of the cellulose from hydrophilic to hydrophobic (*Biliuta & Coseri, 2019*), and an epoxy resin system can be created with better mechanical properties of regular composite materials (*Lu, Askeland & Drzal, 2008*). Results showed that banana fiber modified by alkaline treatment is a low-cost alternative for metal removal in aqueous industry effluents (*Barreto et al., 2010*). Lignin is mostly used as a filler or additive. It may be an excellent candidate for chemical modifications and reactions due to its phenolic and aliphatic hydroxyl groups for the development of new biobased materials (*Laurichesse & Avérous, 2014*; *Bertella & Luterbacher, 2020*).

*Zeng et al. (2014)* reviewed the recent advances in understanding lignin structure in plant cell walls and the negative roles of lignin in the processes of converting biomass to biofuels. Modification of the degree of 4-*O*-methylation of secondary plant cell wall glucuronoxylan is important for better utilization of plant biomass for biofuel production (*Yuan et al., 2014*). *Huang et al. (2019b)* indicated that to enhance biomethane yield of corn stover, *Ceriporiopsis subvermispora* modification is one of the most effective methods. Increasing the number of reducing ends in β-1,4-glucan chains positively affected biomass enzymatic saccharification. This demonstrates a potential strategy for genetic modification of cellulose microfibrils in bioenergy crops (*Huang et al., 2019b*).

## Application of new and special enzymes

Lignin-degrading enzymes that are produced mainly by fungi, are capable of depolymerization and modification of lignin. The most studied representatives of these enzymes are lignin peroxidase, manganese peroxidase and laccase (*Eriksson & Bermek, 2009*; *Zhao et al., 2020*). Dashtban describes ligninolytic enzyme families that are involved

in wood decay processes. The molecular structures, biochemical properties, and the mechanisms of action which make them useful candidates in biotechnological applications, such as pulp bio-bleaching, biosensors, food industries, textile industries, soil bioremediation and in the production of complex polymers in synthetic chemistry are also described (*Dashtban et al., 2010*). With anaerobic digestion of organic waste high in lignocellulose content, methane was synthesized by using laccase, hemicellulose, and cellulase enzymes (*Luo et al., 2010*; *Abraham et al., 2020*).

It was proved that horticultural waste was a potential feedstock for fuel using *Saccharomyces cerevisiae* ethanol production and a special pre-treatment method was developed (*Geng, Xin & Ip, 2012*). Polysaccharide conversion within the lignocellulosic biomass by enzymes is under intensive research (*Horn et al., 2012*). Other researchers isolated a special pectate lyase enzyme that had high thermal and pH stability and was able to produce pectin oligosaccharides from food waste (*Wang et al., 2020c*).

*Li et al. (2018)* investigated the microbial communities involved in anaerobic digestion along with the methane production characteristics of cellulose, hemicellulose, and lignin. The results showed that the biomethane potential of cellulose was higher than that of hemicellulose; however, hemicellulose was hydrolyzed more quickly than cellulose, while lignin was very difficult to be digested (*Li et al., 2018*). Lignocellulosic enzymes, such as cellulases and xylanases were used to modify paper pulp characteristics (*Siqueira et al., 2020*). The tear strength of recycled paper was found to be increased after enzymatic treatment (*Kumar et al., 2018*).

When a commercial cellulase mixture was supplemented with xylanases, substrates could be readily hydrolyzed, recovering most of the hemicellulose and cellulose as monomeric sugars (*Wu, Chandra & Saddler, 2019*). *Ying et al. (2018)* concluded that lignin could bind to cellulases and decrease the enzyme accessibility to cellulose fibers. Modified lignins were prepared from alkali lignin; the post-treatment weakened the adsorption rate between lignin and cellulase enzymes and improved the enzymatic saccharification efficiency. Lignin peroxidase (LiPH8) from the white-rot fungus *Phanerochaete chrysosporium* can efficiently catalyze the degradation of lignin (*Majeke et al., 2020*). The in silico designed and engineered LiPH8 showed improved stability, as well as a higher activity under acidic pH conditions (*Pham et al., 2018*). Several methods have been developed to immobilize cellulase enzymes. The results indicate that cheap supports can be used to effectively immobilize enzymes (*Zang et al., 2018*).

A research group tried the hydrolysis of hemicellulose and cellulose into fermentable sugars which needs a pre-treatment of the whole material to remove unnecessary components from the biomass (*Laca, Laca & Díaz, 2019*). Pentose sugars, which are formed by hydrolyzing hemicelluloses, cannot be fermented with *Saccharomyces cerevisiae*, therefore, other ethanologenic species, for example, *Zymomonas mobilis* are used to break down pentose. *Liu et al. (2019a)* critically reviewed the progress in this field and highlighted challenges and strategies for solutions. An integrated process of cellulase production and pre-treatment employing *Piptoporus betulinus* was proposed to accomplish the economic feasibility of lignocellulosic ethanol (*Li et al., 2019*). Biofuel

produced from lignocellulosic biomass by hydrolysis of fungal consortium enzymes is an environmentally friendly method (*Cherukuri & Akkina, 2019*).

*Wierzbicki et al. (2019)* summarizes biotechnological approaches to improve industrial processing. Structurally different glycosyl hydrolases (enzyme families) were studied. The softwood fiber modification rate by enzymes was studied, dissolved sugar concentrations were measured, and the results were summarized (*Rahikainen et al., 2019*). To achieve an efficient bioethanol production process, cellulase enzymes are under consideration as potential biocatalysts (*Verardi et al., 2020*). Shah made an overview of cellulases, specifically covering their utilization and bioconversion from a proteomics, biochemistry, and genetics perspective (*Shah, Ranawat & Mishra, 2019*).

## Modify the enzymes and increase their efficiency
### Modification and utilization of enzymes by molecular biological, biotechnological, and computational methods

The substrate specificity and activity of enzymes should be increased to reduce the costs of biofuel production. Various qualities can be improved in enzymes, including specificity, catalytic activity and thermostability, all under industrial conditions (*Prajapati et al., 2018*).

In the past few years, cellulases have been successfully immobilized with various methods (*Mubarak et al., 2014*; *Mo et al., 2020*). Cellulase immobilization onto coffee surface provides an excellent base for increasing the enzyme productivity and different uses of the enzyme (*Buntić et al., 2018*). *Gao et al. (2018)* developed a rapid method to immobilize cellulases onto graphene oxide. The immobilization was highly efficient and shows great potential for the immobilization of other enzymes.

Various fungi which break down lignin by their oxidoreductase enzymes (manganese peroxidases, laccases, lignin peroxidases) (*Trametes*, *Polyporus*, *Pleurotus*) can also be used to release cellulose. These enzymes may attack the cellulose and hemicelluloses, so sugar loss can occur during the pre-treatment, and the treatments are very time-consuming (*Houfani et al., 2020*). Fungal enzymes make up more than half of the enzymes currently used in industrial applications. They play a major role in the modification, partial or complete degradation of polysaccharides into oligomers and monomers (*De Vries et al., 2020*).

Cellulase adsorption and inactivation in the presence of lignin is the main concern in the case of lignocellulose conversion to biofuels. Relatively little research deals with the necessary protein structure-function relationships. By molecular modeling, highly active, low lignin-binding cellulases can be determined (*Haarmeyer et al., 2017*).

Understanding enzyme mechanisms at the molecular level is a fundamental requirement in the case of cellulases and ligninases to produce fuel from renewable biomass more efficiently and to utilize plant biomass as a sustainable energy resource. To study the three-dimensional structure of the enzymes and the substrates, along with the enzyme-substrate interactions, computational approaches like molecular dynamics (MD) simulations, ligand-enzyme docking, and quantum chemical calculations are commonly used to give a more detailed description of interaction mechanisms between the enzyme and its substrate (*Meng et al., 2011*).

For selected fungal cellulases, the acidic quality was determined by in silico physicochemical characterization. One molecular docking study revealed that the amino acid side chains of cellulases which interact with cellulose may vary, based on the source organism of the enzyme (*Tamboli et al., 2017*).

The hydrolytic activity of cellulases depends mostly on the binding affinities of the enzyme with cellulose (*Krishnaraj et al., 2017*). These results were obtained by molecular docking and this method is useful in industrial biotechnology to design more effective enzyme structures (*Amore et al., 2017*).

The activity of xylanases can be characterized under different conditions. Computational models can help to scale down the experimental costs and save time by identifying enzymes with appropriate activity for scientific and industrial usage (*Ariaeenejad et al., 2018*).

*Hamre et al. (2019)* made a comparison of three different cellulases by thermodynamic signatures of substrate binding.

Based on an in silico study cold-active enzymes have many more advantages compared to mesophilic enzymes. It is concluded that this approach would be valuable for further research and application at the industrial level (*Latip, Hamid & Nordin, 2019*).

The study of *Mahmood et al. (2019)* shows an approach by an in silico analysis to identify hot spot amino acid residues responsible for enhancing the thermostability of enzymes of industrial importance.

Ligninolytic enzymes can mediate lignin waste degradation. The most common ligninolytic enzymes are laccase, lignin peroxidase, and manganese peroxidase. In the study of *Chen et al. (2011)*, lignin binding to these enzymes was determined. The robustness of the binding modes was verified by MD simulations. Residues Ser113, Pro346, Glu460 in laccase, residues Asp183, Arg43, Ala180 in lignin peroxidase and residues Arg177, Arg42, His173 in manganese peroxidase were the most critical in binding of lignin, respectively. Interaction analyses indicated that hydrophobic interaction plays a crucial function in the designation of substrate specificity. This information can be used as a reference for designing enzymes with better lignin-degrading properties.

*Mäkelä et al. (2020)* described the utilization of wood and dried annual plant biomass using *Basidiomycetes*, along with the biotechnological potential of the fungal enzymes for the use of plant biomass.

*Gerini et al. (2003)* investigated the dynamical and structural properties of lignin peroxidase, a heme-containing enzyme with broad substrate specificity, and its Trp171Ala mutant (*Blodig et al., 2001*) in aqueous solution using MD simulations. The conformational changes in the so-called ligand access channel, which structure was described previously by *Poulos et al. (1993)* have been analyzed to investigate possible variations of its size, which is very important in determining the substrate accessibility to the enzymatic active site. The analysis of the MD trajectories also showed significant fluctuations of the residues forming the ligand access channel, where the open and closed states are in constant equilibrium in solution. These movements grant the access of the substrate to the enzyme active site. Steered molecular dynamics (SMD) docking

simulations have shown that the natural substrate of lignin peroxidase, veratrol, can approach the heme edge through the ligand access channel of the enzyme.

Furthermore, MD simulations made by *Ecker & Fülöp (2018)* showed that small-sized ligands can not necessarily enter the channel due to their specific interactions with channel residues.

To determine the energetics and extent of the interactions between various solvents and the cellulosic oligomers, a force field for ionic liquids was developed by *Liu et al. (2010)*. The behavior of cellulose was examined using MD simulations of a series of glucose oligomers. The simulation included cellulose oligomers in two solvents—methanol and water—which can precipitate cellulose from ionic solutions. The results showed that some of the cations were found to be in strong contact through hydrophobic interactions with the polysaccharides, and the interaction energy between the liquid and the polysaccharide chain was stronger than in the case of methanol or water. These results suggest that cationic interactions play a dominant role in the dissolution of cellulose.

In his article, *Kun et al. (2019)* examined the degradation of plant biomass in a special way, for example, CRISPR/Cas9 genome editing and adaptive evolution.

Similarly, *Jaeger, Burney & Pfaendtner (2015)* used MD to investigate the differences in ionic liquid tolerance among three family 5 cellulases from *Trichoderma viride*, *Thermogata maritima*, and *Pyrococcus horikoshii*. Their analysis demonstrates that the effects of ionic liquids on the enzymes vary in each case from local structural disturbances to loss of much of one of the enzyme's secondary structure. Enzymes with more negatively charged surfaces are much more difficult to destabilize by ionic liquids. Ions in aqueous solutions affect the specificity of enzymes.

*Wohlert & Berglund (2011)* have built a coarse-grained model of cellulose for MD simulations, using cellobiose from the MARTINI coarse-grained force field (*López et al., 2009*). The model was extrapolated to longer cellooligomers, and they obtained a model which gives an ordered structure of cellulose in water.

By mutating tryptophans to alanine, *Payne et al. (2011)* calculated the relative ligand binding free energy of family 6 cellulases (Cel6A). Their results propose that aromatic residues directly upstream of the active site are not directly implied in binding. However, this is necessary for the glucopyranose ring distortion which plays a role in catalysis. Removal of aromatic residues at the entrance and exit of the enzyme tunnel highly affects the binding affinity. This suggests that these residues play a role in chain acquisition and product stabilization. By MD and normal mode analysis, the roles suggested from differences in binding affinity were confirmed. Aromatic-carbohydrate interactions depend on the residue positions in the enzyme channel. These results have implications for understanding protein structure-function relationships in carbohydrate metabolism and protein engineering strategies for biomass utilization since aromatic-carbohydrate interactions are present in all carbohydrate-active enzymes.

*Wakai et al. (2019)* constructed a genetically modified strain that simultaneously contained cellobiohydrolase, endoglucanase, and β-glucosidase; multiple copies of the coding genes were integrated. The resulting transformant showed 40-fold higher cellulolytic activity.

High temperatures are optimal for thermophile cellulases for cleaving sugars from cellulose as was described by *Batista et al. (2011)*. In the thermo-resistant cellulase (Cel9A-68), the cellulose-binding domain (CBM) and the catalytic domain are connected by a Pro-Ser-Thr rich linker. There is a cooperative connection between these domains. The mechanism of action of the CBM is still lacking. Higher collective motions were detected at higher temperatures by simulations. Analysis of the motions showed an interdomain where Gly460 and Asp459, located at the linker region, are the hinge residues. Therefore, thermophile cellulases are useful models to study the interactions of the two domains since these collective motions and cooperative connection need higher temperatures to occur at a detectable scale.

To examine the linker function, *Payne et al. (2013)* performed MD simulations on Family 6 and 7 cellobiohydrolases bound to cellulose. The presence of glycosylated linkers bound to cellulose suggests a previously unknown role in enzyme action. By measuring the binding affinity of the Cel7A CBM and the natively glycosylated Cel7A CBM-linker, this prediction was examined experimentally. The glycosylated linker on the crystalline cellulose increases the affinity of CBM binding. The bound linker may affect enzyme action due to significant damping in the enzyme fluctuations, as MD simulations suggest. The glycosylated linkers in carbohydrate-active enzymes aid in dynamic binding during the enzymatic breakdown of plant cell walls.

Based on the previous study, *Costa, Silva & Batista (2018)* identified two residues (Gly460 and Pro461) located at the linker that act as a hinge point and assumed that a Pro461Gly and a Gly460+ (with an extra glycine) constructs of a Cel9A-68 would present enhanced interdomain motions, while the Gly460Pro mutant would be rigid. The Pro461Gly mutation resulted in an extension of the conformational space, as confirmed by clustering and free energy analyses. If the enzyme has more possible conformations, its substrate-binding and degrading potential may also be higher. The engineering seems to be an efficient way to change the enzyme activity and to facilitate the disruption of cellulose fibers.

The product site of cellulase tunnel is inhibited by monosaccharides and disaccharides, so it is a key ingredient of cellulase action on cellulose. The absolute binding free energy of cellobiose and glucose to the product site of the catalytic tunnel of the Family 7 cellobiohydrolase (Cel7A) was calculated by *Bu et al. (2011)* using two different approaches: SMD simulations and alchemical free energy perturbation molecular dynamics (FEP/MD) simulations. Analysis of the SMD pulling trajectories suggests that several protein residues play key roles in glucose and cellobiose binding to the catalytic tunnel. Five mutations were made computationally to measure the changes in free energy during the product expulsion process. The results showed that new enzymes can make the conversion of biomass more efficiently.

In wood-fed animals, lignocellulose-rich biomass is utilized appropriately by symbiotic microorganisms. Lignocellulose-based biotechnological processes work in a similar way (*Ozbayram, Kleinsteuber & Nikolausz, 2020*).

*Textor et al. (2013)* reported the first crystal structure of the catalytic core domain of Cel7A (cellobiohydrolase I) from *Trichoderma harzianum* IOC 3844. The flexibility of

Tyr side chains in the active center is increased compared to the reference *Trichoderma reesei* Cel7A due to slightly shorter side chains of adjacent amino acids. This creates an additional gap at the side face of the catalytic tunnel. *T. harzianum cellobiohydrolase I interacts with the substrate of T. reesei Cel7A.*

*Silveira & Skaf (2015)* used MD simulations to collect data about the binding of cellobiose to *Trichoderma reesei* Cel7A (TrCel7A) cellobiohydrolase and the effects of mutations that may reduce cellobiose binding, without affecting the integrities of the enzyme. The product binding site shows inside flexibility, that can sterically prevent cellobiose release. The enzyme-substrate interactions can be reduced by point mutations, and not necessarily preserve the structure of the enzyme. Mutation of charged residues in the TrCel7A product binding site causes perturbations that affect the structure of the substrate-binding tunnel and may affect TrCel7A function in other steps of the hydrolysis mechanism. The results show there is a connection between product inhibition and catalytic performance, and these may designate directions for cellulase engineering.

### Modification of enzymes using molecular modeling methods

There are several ways to improve the enzymes currently used for biomass processing. Molecular modeling techniques can be used to modify enzymes to function more efficiently under the given physicochemical conditions.

By studying the function of the enzymes and catalytic conditions, it is possible to intervene in the chemical reactions by modifying the enzyme structure to better suit the actual processing. To do this, we need to know the enzymatic processes used and modify the function of enzymes by computer modeling and design, in the hope of better utilization.

Nonspecific adsorption of cellulases to lignin prevents enzymatic biomass conversion. Computations show that cellulase with a negatively supercharged surface could reduce lignin inhibition. It is possible to construct highly active cellulases that are resistant to lignin-mediated inactivation, although further work is needed to understand this problem (*Whitehead et al., 2017*).

Enzymes that break down polysaccharides are often linked with glycosylation, *N*- and *O*-linked glycans, the roles of which are only partially understood. Glycans may affect the critical properties of enzymes: *N*-glycosylation improves thermal and proteolytic stability, *O*-glycosylation improves CBM-binding affinity and stability along with proteolytic stability, but their presence not necessarily affect catalytic activity (*Amore et al., 2017*). Modeling of glycosylated cellulases can improve our knowledge about the various functions of the glycans.

*Chung et al. (2019)* described a glycosylated cellulase where mainly galactose disaccharides could be found. This glycosylation dramatically impacted the hydrolysis of insoluble substrates, proteolytic and thermal stability and was found to be necessary for this enzyme to function in harsh environments including industrial settings.

It is of note that immobilized cellulase-polymethacrylate particles exhibit excellent pH-adaptability compared to free cellulase (*Chan et al., 2019*).

## CONCLUSIONS

We have explored the possibilities of plant structural biomass utilization along with methods recently studied and applied. Progress has been made in all areas and this field is in constant evolution. New physical and chemical methods have been developed for the pre-treatment of biomass, which can result in extracting biomass materials more efficiently. A wide variety of enzymes in microorganisms have been isolated and used in biomass processing. The mechanism of enzyme function has been studied and new and modified methods have been developed. In our view, there are a few areas that have been seen less development recently. These include changing the properties of existing enzymes using rational design based on structure modeling, MD simulations, enzyme substrate interactions, and virtual mutagenesis.

There are untapped and enormous opportunities to produce modified enzymes that work more effectively under the usually harsh industrial conditions used for biomass processing. Using rational modification of enzymes, the development of biomass conversion, enzyme or process effectiveness can be increased significantly. Physical, chemical, biochemical, biotechnological and molecular biological methods are available to modify the protein side chains in enzymes to provide more efficient biomass utilization that are more appropriate to the techniques employed.

With the widespread application of these methods, further progress is expected in this field, and it can only be supported that in silico modeling, virtual modification of molecules, molecular simulations are used to explore enzyme action under various environmental and/or industrial conditions. This would help to develop the industrial biomass utilization methods used so far without having to change the method, but only to modify the specific enzymes to work more efficiently under the given conditions, thus increasing productivity and reducing costs.

## ACKNOWLEDGEMENTS

The authors thank Dr. Tamás Ponyi and Csaba Biegl PhD for their helpful linguistic and professional comments.

### Funding

The authors received no funding for this work.

### Competing Interests

The authors declare that they have no competing interests.

### Author Contributions

- László Fülöp conceived and designed the experiments, performed the experiments, analyzed the data, authored or reviewed drafts of the paper, and approved the final draft.
- János Ecker conceived and designed the experiments, performed the experiments, analyzed the data, authored or reviewed drafts of the paper, and approved the final draft.

## Data Availability

We do not have raw data for this literature review.

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
