# Peer review of "An overview of biomass conversion: exploring new opportunities"

_PeerJ, doi:10.7717/peerj.9586_

## Round 0.1 · original submission · Major Revisions

Please consider following things while revising your manuscript:
- Overall language/grammar needs some help, and the structure/flow of information is not smooth - you need to organize it properly - adding your hypothesis - backed by recent publications.
- Introduction should be edited, and overall add/cite/discuss more recent and relevant references throughout manuscript, as much work is reported in this exciting area of research (more specifically enzyme research).
- Please also modify your figures, in such a way that it is relevant to your data/manuscript.
- Please write conclusions identifying unresolved questions/gaps and future directions in the research field.
For further details, please see reviewers' comments, kindly edit your manuscript accordingly.

·

Basic reporting

This paper intended to overview the plant biomass processing methods and intend to explore the possibilities of new methods and enzymes to be used in biomass recyclization. But the structure and logic of the whole manuscript is not very clear, and the authors should have thought more deeply and provide more own perspective and important conclusions.

Experimental design

no comment

Validity of the findings

no comment

Additional comments

This paper intended to overview the plant biomass processing methods and intend to explore the possibilities of new methods and enzymes to be used in biomass recyclization. But the structure and logic of the whole manuscript is not very clear, and the authors should have thought more deeply and provide more own perspective and important conclusions.
1. Line 22-24, actually, lots of work has been done in the modification of enzymes and alter the function of enzymes.
2. Your introduction needs more detail.
3. Line 390-408, Enzyme modification, in this part, the author should review the progress on enzyme modification, especially advances in methods, great progress have been made in recent years.
4. The references should be arranged in alphabetical order.

Reviewer 2 ·

Basic reporting

Must be improved - see comments for suggestions. Notably, the structure of the manuscript lacks coherence and the figure has no strong contribution to the goal of this review.

Literature references are not sufficient for lines before Line 109.

Experimental design

no comment.

Validity of the findings

no comment.

Additional comments

The manuscript intends to review "the results of plant biomass processing methods", including pretreatment methods (2 pages), enzymes for biomass processing (1.5 pages), and optimization of enzymes (computational modeling for 3 pages and experimental confirmation for 0.5 pages). The basic concepts might be useful, however they are marred by poor presentation. The focus of the manuscript seems to be shifted to computation modeling later on without enough rationals.

Line 31. Can the authors provide more recent data?
Line 51. What's the significance of mentioning "Hydrolysis of the ...... molecules?
Line 58. Lack of coherence: The authors mentioned the hydrolysis of hemicelluloses takes place under milder condition. However, the next sentence goes to the hydrolysis conditions for cellulose and lost track of hemicellulose conditions.
Line 64. The sentence is the same with the previous one.
Line 69-72. What's the significance of mentioning "pectin"? It was never mentioned again later on. If this is part of the introduction, the authors should focus on the topics that are useful for main results/discussion.
Line 31-109. Lack of literature supports.
Line 113. check grammar.
Line 130. Add reference.
Line 131. What does "this purpose" refer to?
Line 135. "Modified micro fibrillated cellulose was used" for what purpose?
Line 137. "changed the surface character" of which component?
Line 135-143. The paragraph before and after this seem to be talking about pre-treatment. However, these lines seem to be talking about modification? Should move this paragraph to somewhere else.
Line 144-146. Why is this sentence relevant to the topic and the context?
Line 150-153. Why is this sentence relevant to the topic and the context?
Line 157-159. "...elasticity..." Why is this sentence relevant to the topic and the context?
Line 160. "....the reduction" of which component?
Line 163. Why is this sentence relevant to the topic and the context?
LIne 164-170. this paragraph seems to be sentences of information pasted together with no connection to each other.
Line 171-173. Why is this sentence relevant to the topic and the context?
Line 187-189. This paragraph seems to be talking about modification? but this sentence is not about modification. It does not go well with the context.
Line 197-200. It's important to state what other researcher has done. But the authors provided methods instead of results, showing no relevance to the context or the manuscript.
Line 207. Why immobilize? This sentence might go better with the next paragraph
Line 219-221. Maybe it's better to lead the paragraph with this sentence.
Line 234. "by enzymes..." Please specify which enzymes.
Line 258. reference.
Line 264-265. Why is this sentence relevant to the topic and the context?
Line 281-281. This sentence is a repeat of previous sentence.
Line 289-291. Why is this sentence relevant to the topic and the context? What results did the reference show?
Line 303. Why is this figure relevant to the manuscript?
Line 306-309. Why is this relevant to the topic and the context?
Line 342-347. Why is this knowledge relevant to the topic?
Line 352. "...broader space..." what does this imply and why is this relevant to the topic?
Line 370. "To examine the linker function..." should this be moved up to Line 354 so it goes together with other "linker" topic?
Line 402. "may affect the critical properties of enzymes" how? and why is this relevant to the topic?

---

## Round 0.2 · accepted · Accept

Authors revised the content considering all comments and the revised manuscript reads much better and more clear/informative now.

Reviewer 2 ·

Basic reporting

no comment

Experimental design

no comment

Validity of the findings

no comment

Additional comments

The coherence of the manuscript was much improved! The introduction is sufficient and the rational of the study is clear. The authors' rebuttal provides some interesting aspects for the significance of "modification of enzymes using molecular modeling methods": "There are very few articles on the latter...All this is possible using special molecular modeling methods... 50 new articles, most of which are from the last two years". Incorporating the rebuttal into main manuscript might help stress the importance and point out the fast evolving nature of this area of research.